# Combinations of alcohol-induced flushing with genetic polymorphisms of alcohol and aldehyde dehydrogenases and the risk of alcohol dependence in Japanese men and women

**Akira Yokoyama**[1]\*, **Tetsuji Yokoyama**[2], **Mitsuru Kimura**[1], **Sachio Matsushita**[1], **Masako Yokoyama**[3]

1 National Hospital Organization Kurihama Medical and Addiction Center, Yokosuka, Kanagawa, Japan,
2 Department of Health Promotion, National Institute of Public Health, Wako, Saitama, Japan, 3 Mitsukoshi Health and Welfare Foundation, Shinjuku-ku, Tokyo, Japan

\* a-yoko@ab.auone-net.jp

## Abstract

### Objective

The risk of alcohol dependence (AD) in Japanese men and women was evaluated according to combinations of alcohol flushing and *aldehyde dehydrogenase-2* (*ALDH2*, rs671) and *alcohol dehydrogenase-1B* (*ADH1B*, rs1229984) genotypes, all of which are known to determine AD susceptibility in Asians. Previous studies have focused on men, since women account for a smaller proportion of AD subjects.

### Methods

Case control studies were conducted between 3721 male and 335 female AD Japanese and 610 male and 406 female controls who were asked about their current or former tendency to experience facial flushing after drinking a glass of beer and underwent *ALDH2* and *ADH1B* genotyping. The time at which alcohol-induced facial flushing tendencies had disappeared in former-flushing AD subjects was also evaluated.

### Results

Current alcohol flushing, the inactive *ALDH2\*1/\*2* genotype, and the fast-metabolizing *ADH1B\*2* allele were less frequently found in the AD groups. Although alcohol flushing was strongly influenced by the *ALDH2* and *ADH1B* genotypes, multiple logistic model showed that never or former flushing and the genotype combinations were independent strong risk factors of AD in men and women. Never or former flushing (vs. current flushing) markedly increased the odds ratios of AD in carriers of each of the *ALDH2* and *ADH1B* genotype combinations. The temporal profiles for drinking and flushing in former-flushing AD subjects

**Data Availability Statement:** All relevant data are within the manuscript and its Supporting information files.

**Funding:** The Authors received no specific funding for this work.

**Competing interests:** The authors have declared that no competing interests exist.

revealed that the flushing response disappeared soon after or before the start of habitual drinking during young adulthood, regardless of the *ALDH2* genotype.

## Conclusion

Although alcohol flushing is influenced by the *ALDH2* and *ADH1B* genotypes, constitutional or acquired flushing tolerance is an independent susceptibility trait for AD. The combination of the alcohol flushing status and the *ALDH2* and *ADH1B* genotypes can provide a better new strategy for AD risk assessment than the alcohol flushing status alone or the genotypes alone in Asian men and women.

## Introduction

Alcohol dehydrogenases (ADHs) oxidize ethanol to acetaldehyde, and then aldehyde dehydrogenases (ALDHs) oxidize acetaldehyde to acetate. The inactive *ALDH2*2* allele (rs671), which results in slow acetaldehyde elimination, and the fast-metabolizing *ADH1B*2* allele (rs1229984), which results in fast acetaldehyde production, both enhance alcohol-induced facial flushing [1–6] and are strong Asian protective factors against alcohol dependence [7–10]. *ALDH2* plus *ADH1B* genotyping using buccal smear DNA is commercially available and inexpensive in Japan. Knowing an individual's own *ALDH2* plus *ADH1B* genotype is a growing preventive strategy against alcohol dependence.

However, alcohol-dependent individuals generally experience a weak or absent alcohol-induced facial flushing tendency, regardless of their *ALDH2* and *ADH1B* genotypes; for example, 86.4% of Japanese alcohol-dependent men with the inactive heterozygous *ALDH2*1/*2* genotype did not report current flushing and instead reported never or former flushing [6]. We empirically noticed that patients with alcohol dependence who were never or former flushers were often surprised to learn that they had inactive ALDH2, because many never flushers had believed that they had active ALDH2 and many former flushers had experienced alcohol flushing for only short periods of time during their young adulthood [6]. The combination of the alcohol flushing status and *ALDH2* plus *ADH1B* genotype may have a better performance for predicting the alcohol-dependence risk in comparison with alcohol flushing alone or genotyping alone.

The time at which an alcohol-induced facial flushing tendency disappeared in former-flushing, alcohol-dependent subjects should also be evaluated. Most of the research in this field has focused on alcohol-dependent men, since women accounted for a very small proportion of alcohol-dependent patients among Asians [11].

According to a report, alcohol consumption in inactive *ALDH2*1/*2* carriers in the general population increased in the order of current flushers, former flushers, and never flushers [12]. If the flushing disappearance occurred in young adulthood and not around the time of development of alcohol dependence in many former flushers who are alcohol-dependent, it is conceivable that both former flushing and never flushing may serve as independent risk factors for alcohol dependence, in combination with the *ADH1B* and *ALDH2* genotypes.

To evaluate these issues, we compared a large dataset of Japanese alcohol-dependent men and women with historical control groups of Japanese men and women who had undergone annual health checkups. In addition, we evaluated the temporal profiles of drinking and flushing in former-flushers with alcohol dependence.

## Materials and methods

### Subjects

The study subjects were 3721 Japanese men and 335 Japanese women aged 40–79 years with alcohol dependence who met the following criteria: (a) visited the Kurihama Medical and Addiction Center for treatment of alcohol dependence for the first time between 2004 and 2017 in men and between 2006 and 2017 in women, (b) were asked about their alcohol flushing, and (c) underwent routine *ADH1B* and *ALDH2* genotyping. The questions were asked with both the questioners and the patients having no knowledge of the patients' *ADH1B* and *ALDH2* genotypes. All of the subjects met the ICD-10 or DSM-IV criteria for alcohol dependence [13, 14]. Male subjects with alcohol dependence were a subgroup (40–79 years) of 4051 subjects aged 30–79 years, in which we recently reported the associations between the results of an alcohol flushing questionnaire and the *ALDH1B* and *ALDH2* genotypes [6]. We borrowed historical control data of 1016 Japanese subjects (610 men and 406 women) aged 40–79 years who had visited two Tokyo clinics for annual health checkups and had registered in case control studies of esophageal cancer as cancer-free controls between 2000 and 2001 [12, 15]. Most of the controls were ordinary residents or workers lived in Tokyo or neighboring areas. Kanagawa, where our center is located, is situated next to Tokyo, and previous studies have shown similar *ADH1B* and *ALDH2* genotype distributions in Kanagawa and Tokyo [12, 15]. Although alcohol-dependent individuals had not been excluded from the controls, the lifetime frequency of alcohol dependence based on ICD-10 criteria has been estimated to be only 1.5% in men and 0.2% in women among Japanese adults according to 2003 nationwide survey [11]. The anonymous historical control data consisted of age, results of the flushing questionnaire, and the *ADH1B* and *ALDH2* genotypes.

Between July 2019 and December 2020, 283 Japanese subjects with alcohol dependence (253 men and 30 women) aged 40–79 years who visited the Center for the first time were asked about their alcohol flushing history and underwent routine *ADH1B* and *ALDH2* genotyping. Of the subjects, 7 (6 men and one women), 69 (64 men and 5 women), and 207 (183 men and 24 women) reported themselves as current flushers, former flushers, and never flushers, respectively. We asked 66 former flushers (61 men and 5 women) the following question: "Up until what age did you have a tendency to develop facial flushing immediately after drinking a glass of beer?"

The ethics committee of the Kurihama Medical and Addiction Center reviewed and approved the proposed study, and all procedures involved in this study were performed in accordance with the Declaration of Helsinki with written informed consent from each participant.

### The simple flushing questionnaire

The physicians verbally questioned the alcohol-dependent subjects about flushing using two simple questions [12]: (a) "Do you have a tendency to flush in the face immediately after drinking a glass of beer?" and (b) "Did you have a tendency to flush in the face immediately after drinking a glass of beer during the first 1–2 years after you started drinking?" A "glass" means about 180 ml, the volume of the most common size Japanese beer glass. The term "current flushing" was applied to individuals who answered "yes" to question (a), and the term "former flushing" to those who answered "no" to question (a) and "yes" to question (b). The term "never flushing" was applied to the remaining subjects. The control subjects had completed a self-reported questionnaire consisting of the same questions.

### ALDH2 and ADH1B genotypes

The *ALDH2* and *ADH1B* genotypes were determined using polymerase chain reaction-restriction fragment length polymorphism methods on blood DNA samples obtained from the subjects [16, 17].

### Statistical analyses

Data were summarized as the mean and standard deviation (SD) or the percentage values and compared between alcohol dependence and control groups by Student's *t*-test or $\chi^2$ test, respectively. Age-adjusted percentage values were calculated by the direct method and statistically tested by the Cochran-Mantel-Haenszel test. Multivariate odds ratios (ORs) and the 95% confidence intervals (CIs) were estimated using multiple logistic models. A *p* value of <0.05 was considered as statistically significant. All the statistical analyses were performed using the SAS statistical analysis software (version 9.4; SAS Institute, Cary, NC).

## Results

Table 1 shows the age distribution, the results of the simple flushing questionnaire, and the *ALDH2* and *ADH1B* genotype combinations in men and women, respectively. Since

**Table 1. Alcohol flushing and *ADH1B* and *ALDH2* genotypes in the male and female subjects.**

| | | Men | | | | P | Women | | | | P |
|---|---|---|---|---|---|---|---|---|---|---|---|
| | | Alcohol dependence (n = 3721) | | Controls (n = 610) | | | Alcohol dependence (n = 335) | | Controls (n = 406) | | |
| | | n | % | N | % | | n | % | n | % | |
| Age | | | | | | | | | | | |
| 40–49 | | 1077 | 28.9% | 41 | 6.7% | <0.0001[a] | 155 | 46.3% | 29 | 7.1% | <0.0001 [a] |
| 50–59 | | 1266 | 34.0% | 285 | 46.7% | | 110 | 32.8% | 202 | 49.8% | |
| 60–69 | | 983 | 26.4% | 254 | 41.6% | | 56 | 16.7% | 144 | 35.5% | |
| 70–79 | | 395 | 10.6% | 30 | 4.9% | | 14 | 4.2% | 31 | 7.6% | |
| Mean±SD | | 56.2±9.7 | | 58.9±7.1 | | <0.0001[b] | 52.2±9.0 | | 58.7±7.6 | | <0.0001[b] |
| Alcohol flushing | | | | | | | | | | | |
| Never flushing | | 3037 | 81.6% | 315 | 51.6%* | <0.0001[c] | 296 | 88.4% | 226 | 55.7%* | <0.0001[c] |
| Former flushing | | 553 | 14.9% | 68 | 11.1%* | | 31 | 9.3% | 26 | 6.4% | |
| Current flushing | | 131 | 3.5% | 227 | 37.2%* | | 8 | 2.4% | 154 | 37.9%* | |
| *ALDH2* genotype | *ADH1B* genotype | | | | | | | | | | |
| *1/*1 (active) | *1/*1 (slow) | 845 | 22.7% | 11 | 1.8%* | <0.0001[c] | 87 | 26.0% | 13 | 3.2%* | <0.0001[c] |
| | *1/*2 (fast) | 1061 | 28.5% | 121 | 19.8%* | | 104 | 31.0% | 75 | 18.5%* | |
| | *2/*2 (fast) | 1215 | 32.7% | 194 | 31.8% | | 102 | 30.4% | 133 | 32.8% | |
| *1/*2 (inactive) | *1/*1 (slow) | 194 | 5.2% | 17 | 2.8%* | | 18 | 5.4% | 13 | 3.2% | |
| | *1/*2 (fast) | 181 | 4.9% | 78 | 12.8%* | | 8 | 2.4% | 63 | 15.5%* | |
| | *2/*2 (fast) | 225 | 6.0% | 147 | 24.1%* | | 16 | 4.8% | 87 | 21.4%* | |
| *2/*2 (inactive) | Any | 0 | 0.0% | 42 | 6.9%* | | 0 | 0.0% | 22 | 5.4%* | |

*ADH1B*, alcohol dehydrogenase-1B; *ALDH2*, aldehyde dehydrogenase-2

[a] $\chi^2$ test;

[b] Student's *t*-test;

[c] Cochran-Mantel-Haenszel test adjusted for age.

* *P* <0.05 for comparing the prevalence of each category between alcohol dependence and controls. [c]

significantly different age distribution was observed between the alcohol-dependent groups and the controls, the subsequent multivariate analyses were performed with age-adjustment. Striking differences in the results of the flushing questionnaire and the *ALDH2* and *ADH1B* genotypes were shown between the alcohol-dependent groups and the controls regardless of gender. The inactive *ALDH2*2/*2* homozygotes were not found in the alcohol-dependent groups, while in 6.9% and 5.4% of the male and female controls, respectively. Current alcohol flushing, the inactive *ALDH2*1/*2* genotype, and the fast-metabolizing *ADH1B*2* allele were less frequently found similarly in the alcohol-dependent men and women groups.

Figs 1 and 2 show the comparison of the alcohol flushing according to the combination of *ALDH2* and *ADH1B* genotypes in men and women, respectively. In comparison with the

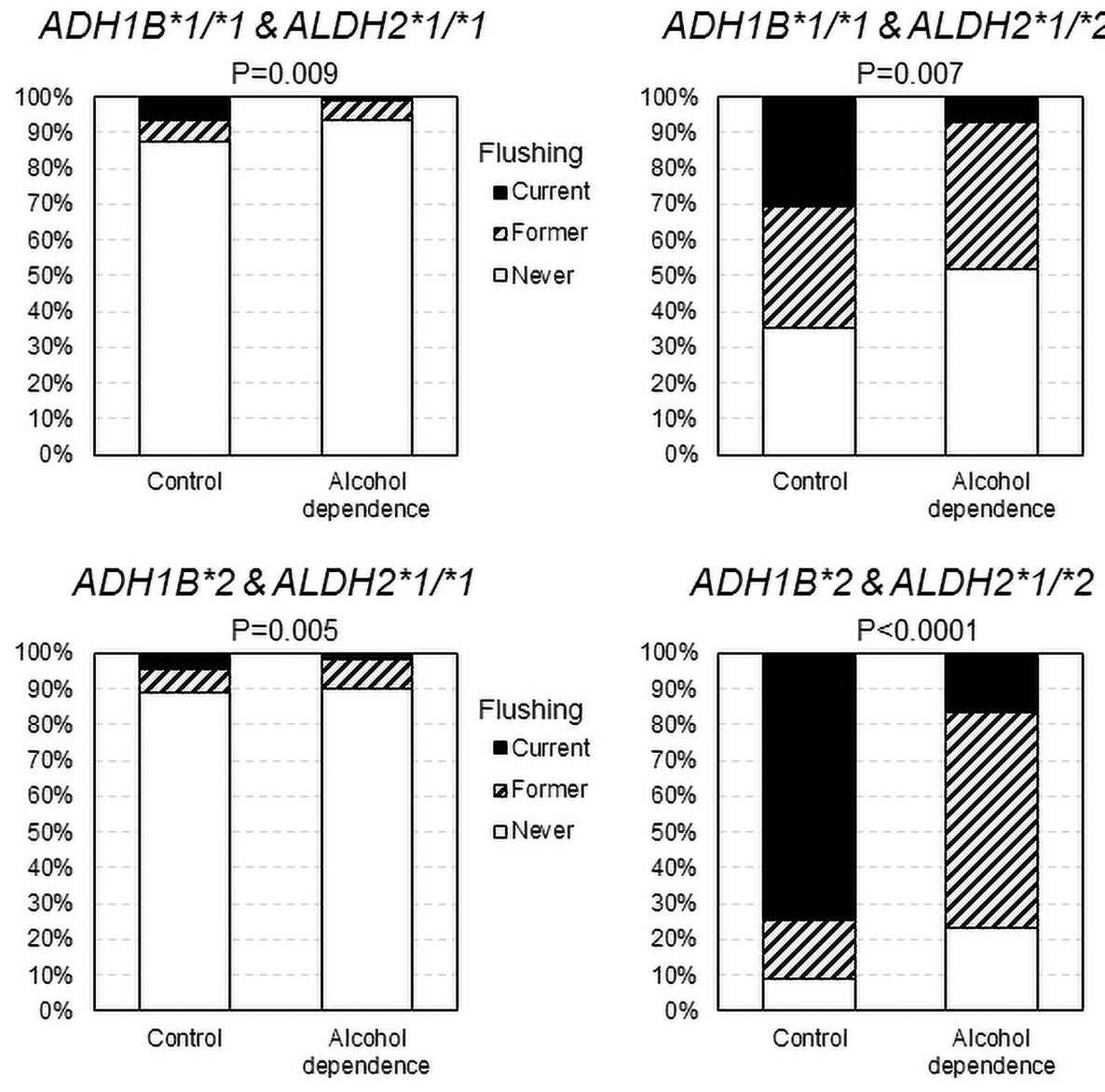

**Fig 1. Comparison of the alcohol flushing between alcohol-dependent men and controls according to the combinations of *ALDH2* and *ADH1B* genotypes.** Never or former flushing was more frequently reported in the alcohol-dependent men than in the controls regardless of the *ALDH2* and *ADH1B* genotype combination. Percentage values were adjusted for age by a direct method using the alcohol dependence group as the standard population; *p* values were by Cochran-Mantel-Haenszel test adjusted for age.

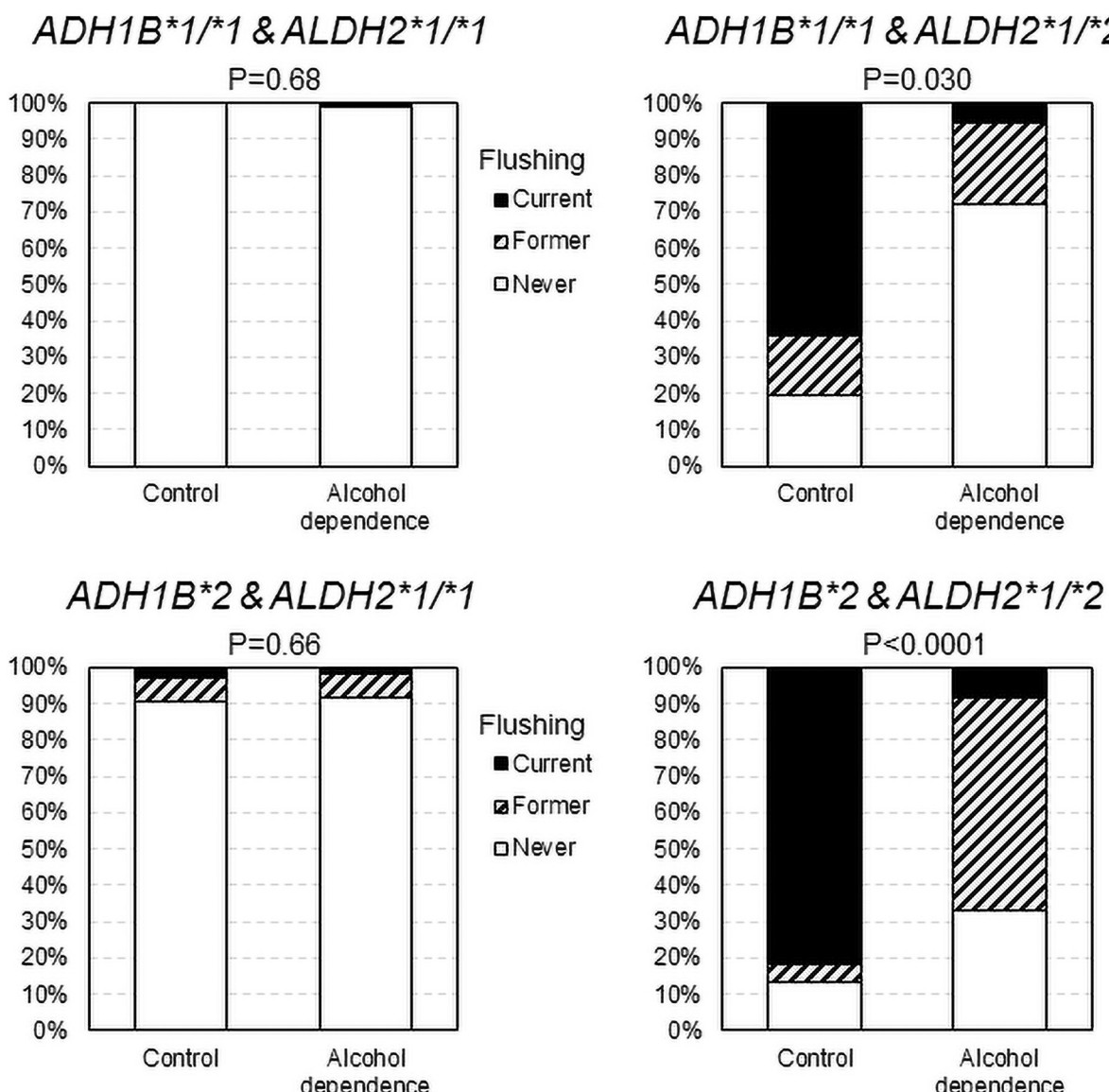

**Fig 2. Comparison of the alcohol flushing between alcohol-dependent women and controls according to the combinations of *ALDH2* and *ADH1B* genotypes.** Never or former flushing was more frequently reported in the alcohol-dependent women than in the controls among the *ALDH2*1/*2* genotype carrier. Percentage values were adjusted for age by a direct method using the alcohol dependence group as the standard population; *p* values were by Cochran-Mantel-Haenszel test adjusted for age.

controls, never or former flushing was more frequently reported in the alcohol-dependent men with any combinations of the *ADH1B* and *ALDH2* genotypes, and in the alcohol-dependent women with the *ALDH2*1/*2* genotype.

Tables 2 and 3 show the age-adjusted ORs and 95% CIs of alcohol dependence according to the reported alcohol flushing status and combinations of the *ALDH2* and *ADH1B* genotypes using multiple logistic regression models in men and women, respectively. When current flushing was used as a reference, never flushing (OR [95% CI] = 18.1 [13.9–23.5] in men and 21.2 [9.89–45.6] in women) and former flushing (15.5 [10.9–22.0] in men and 19.2 [7.52–49.0] in women) markedly increased the risk of alcohol dependence. When the *ALDH2*1/*2* and *ADH1B*2/*2* combination was used as a reference, the increased number of *ALDH2*1* allele

**Table 2. Age-adjusted odds ratios (95% confidence intervals) of alcohol dependence according to alcohol flushing status and/or the combinations of *ALDH2* and *ADH1B* genotypes in the male subjects.**

| | | Age-adjusted OR (95%CI) of alcohol dependence | | | |
| --- | --- | --- | --- | --- | --- |
| | | Number of AD/control | Flushing alone | Genotype alone | Flushing and Genotype |
| Alcohol flushing | | | | | |
| Never flushing | | 3037/315 | 18.1 (13.9–23.5) | - | 7.30 (5.13–10.4) |
| Former flushing | | 553/68 | 15.5 (10.9–22.0) | - | 10.6 (7.37–15.3) |
| Current flushing | | 131/227 | 1 (ref.) | - | 1 (ref.) |
| *ALDH2* genotype | *ADH1B* genotype | | | | |
| *1/*1 (active) | *1/*1 (slow) | 845/11 | - | 46.7 (24.8–88.2) | 23.6 (11.9–46.8) |
| | *1/*2 (fast) | 1061/121 | - | 5.64 (4.22–7.53) | 2.88 (1.97–4.21) |
| | *2/*2 (fast) | 1215/194 | - | 4.00 (3.07–5.23) | 2.05 (1.42–2.95) |
| *1/*2 (inactive) | *1/*1 (slow) | 194/17 | - | 7.24 (4.19–12.5) | 4.34 (2.41–7.82) |
| | *1/*2 (fast) | 181/78 | - | 1.42 (1.00–2.01) | 1.27 (0.85–1.90) |
| | *2/*2 (fast) | 225/147 | - | 1 (ref.) | 1 (ref.) |
| *2/*2 (inactive) | Any | 0/42 | - | 0 (NA) | 0 (NA) |

*ADH1B*, alcohol dehydrogenase-1B; *ALDH2*, aldehyde dehydrogenase-2; NA, not applicable

Odds ratio (OR) and 95% confidence interval (CI) were estimated by a logistic regression model.

- Not included in the model.

and *ADH1B*\*1 allele in the genotype combinations resulted in the higher ORs of alcohol dependence, and the OR (95%CI) by the *ADH1B*\*1/\*1 and *ALDH2*\*1/\*1 combination was the highest (46.7 [24.8–88.2] in men and 31.8 [13.7–73.8] in women). A multivariate model including both the alcohol flushing status and the *ALDH2* and *ADH1B* genotype combinations showed that the alcohol flushing status and the *ADH1B* and *ALDH2* genotype combinations were strong independent strong risk factors of alcohol dependence in both men and women,

**Table 3. Age-adjusted odds ratios (95% confidence intervals) of alcohol dependence according to alcohol flushing status and/or the combinations of *ALDH2* and *ADH1B* genotypes in the female subjects.**

| | | Age-adjusted OR (95%CI) of alcohol dependence | | | |
| --- | --- | --- | --- | --- | --- |
| | | Number of AD/control | Flushing alone | Genotype alone | Flushing and genotype |
| Alcohol flushing | | | | | |
| Never flushing | | 296/226 | 21.2 (9.89–45.6) | - | 7.57 (3.04–18.8) |
| Former flushing | | 31/26 | 19.2 (7.52–49.0) | - | 14.0 (5.26–37.5) |
| Current flushing | | 8/154 | 1 (ref.) | - | 1 (ref.) |
| *ALDH2* genotype | *ADH1B* genotype | | | | |
| *1/*1 (active) | *1/*1 (slow) | 87/13 | - | 31.8 (13.7–73.8) | 15.8 (5.84–42.8) |
| | *1/*2 (fast) | 104/75 | - | 6.71 (3.44–13.1) | 3.24 (1.40–7.51) |
| | *2/*2 (fast) | 102/133 | - | 3.66 (1.91–7.00) | 1.78 (0.78–4.04) |
| *1/*2 (inactive) | *1/*1 (slow) | 18/13 | - | 6.79 (2.55–18.1) | 4.85 (1.62–14.6) |
| | *1/*2 (fast) | 8/63 | - | 0.56 (0.21–1.51) | 0.56 (0.18–1.73) |
| | *2/*2 (fast) | 16/87 | - | 1 (ref.) | 1 (ref.) |
| *2/*2 (inactive) | Any | 0/22 | - | 0 (NA) | 0 (NA) |

AD, alcohol dependence; *ADH1B*, alcohol dehydrogenase-1B; *ALDH2*, aldehyde dehydrogenase-2; NA, not applicable

Odds ratio (OR) and 95% confidence interval (CI) were estimated by a logistic regression model.

- Not included in the model.

**Table 4. Age-adjusted odds ratios (95% confidence intervals) of alcohol dependence according to the combinations of alcohol flushing status and *ALDH2* and *ADH1B* genotypes in the male subjects.**

| *ALDH2* genotype | *ADH1B* genotype | Alcohol flushing | | | | | |
|---|---|---|---|---|---|---|---|
| | | Current flushing | | Former flushing | | Never flushing | |
| | | Number of AD/control | OR (95%CI) | Number of AD/control | OR (95%CI) | Number of AD/control | OR (95%CI) |
| *1/*1 (active) | *1/*1 (slow) | 8/1 | 17.0 (1.92–150) | 47/1 | 138 (18.2–999<) | 790/9 | 241 (112–517) |
| | *1/*2 (fast) | 23/4 | 12.9 (4.04–41.5) | 89/3 | 78.0 (23.0–265) | 949/114 | 24.3 (15.7–37.4) |
| | *2/*2 (fast) | 19/11 | 5.88 (2.50–13.9) | 93/19 | 13.0 (6.86–24.6) | 1103/164 | 19.2 (12.6–29.3) |
| *1/*2 (inactive) | *1/*1 (slow) | 14/5 | 7.40 (2.38–23.1) | 79/6 | 42.3 (16.8–106) | 101/6 | 43.1 (17.2–108) |
| | *1/*2 (fast) | 25/57 | 1.12 (0.59–2.10) | 116/12 | 27.5 (13.4–56.2) | 40/9 | 11.2 (4.82–26.0) |
| | *2/*2 (fast) | 42/108 | 1 (ref.) | 129/27 | 14.7 (8.29–26.2) | 54/12 | 13.7 (6.45–28.9) |

AD, alcohol dependence; *ADH1B*, alcohol dehydrogenase-1B; *ALDH2*, aldehyde dehydrogenase-2; NA, not applicable

Age-adjusted odds ratio (OR) and 95% confidence interval (CI) were estimated by a logistic regression model using *ALDH2*1/*2* (inactive) plus *ADH1B*2/*2* (fast) plus current flushing as the reference group.

though each OR in this model was lower than that in a model that included the alcohol flushing status alone or the genotype combinations alone.

Table 4 shows the age-adjusted ORs of alcohol dependence according to the combinations of reported alcohol flushing and *ALDH2* and *ADH1B* genotype combinations in men. When the combination of the current flushing, *ALDH2*1/*2*, and *ADH1B*2/*2* was used as a reference, the never or former flushing was markedly increased the ORs of alcohol dependence in the carriers of each *ALDH2* and *ADH1B* genotype combination. Although the estimated ORs were unstable due to the small number in current flushing category in alcohol-dependent women with each genotype combination, similar results were obtained in women (Table 5).

Table 6 shows the temporal profiles for drinking and flushing in 66 former-flushing, alcohol-dependent subjects according to their *ALDH2* genotypes. No differences in age, age at the start of drinking, age up until which the flushing tendency continued, duration of flushing, age at the start of habitual drinking, and temporal profiles between alcohol flushing and habitual drinking were seen according to *ALDH2* genotype. The mean age up until which the flushing tendency continued, the mean age at the start of habitual drinking, and the mean age at the first hospital visit for alcohol-related problems were 25.7±6.1 years, 25.9±7.4 years, and 51.4 ±12.4 years, respectively. The flushing tendency had disappeared before and within 5 years of the start of habitual drinking in 30 (45.5%) and 24 (36.4%) of the 66 former-flushing subjects, respectively.

## Discussion

The present study showed that the never or former flushing by the simple flushing questionnaire was a strong independent risk factor of alcohol dependence in any *ALDH2* and *ADH1B* genotype combination carriers in both men and women. The increased number of *ALDH2*1* allele and *ADH1B*1* allele in the *ALDH2* and *ADH1B* genotype combinations resulted in the higher risk of alcohol dependence in women as well as men, and similar ORs of alcohol dependence was observed between women and men. Flushing responses after a small amount of

**Table 5. Age-adjusted odds ratios (95% confidence intervals) of alcohol dependence according to the combinations of alcohol flushing status and *ALDH2* and *ADH1B* genotypes in the female subjects.**

| *ALDH2* genotype | *ADH1B* genotype | Alcohol flushing | | | | | |
|---|---|---|---|---|---|---|---|
| | | Current flushing | | Former flushing | | Never flushing | |
| | | Number of AD/ control | OR (95%CI) | Number of AD/ control | OR (95%CI) | Number of AD/ control | OR (95%CI) |
| *1/*1 (active) | *1/*1 (slow) | 1/0 | NA | 0/0 | - | 86/13 | 346 (42.8–999<) |
| | *1/*2 (fast) | 2/2 | 54.8 (2.89–999) | 7/3 | 111 (9.19–999<) | 95/70 | 73.9 (9.74–561) |
| | *2/*2 (fast) | 2/7 | 18.2 (1.32–251) | 6/6 | 68.3 (6.61–704) | 94/120 | 40.6 (5.38–306) |
| *1/*2 (inactive) | *1/*1 (slow) | 1/7 | 7.47 (0.36–153) | 4/2 | 111 (7.30–999<) | 13/4 | 177 (17.3–999<) |
| | *1/*2 (fast) | 1/49 | 1.14 (0.07–19.4) | 5/6 | 33.2 (2.94–374) | 2/8 | 11.6 (0.82–163) |
| | *2/*2 (fast) | 1/68 | 1 (ref.) | 9/9 | 52.6 (5.57–497) | 6/10 | 34.6 (3.51–342) |

AD, alcohol dependence; *ADH1B*, alcohol dehydrogenase-1B; *ALDH2*, aldehyde dehydrogenase-2; NA, not applicable

Age-adjusted odds ratio (OR) and 95% confidence interval (CI) were estimated by a logistic regression model using *ALDH2*1/*2* (inactive) plus *ADH1B*2/*2* (fast) plus current flushing as the reference group.

drinking have been mainly attributable to high acetaldehyde exposure, and are influenced by the *ALDH2* and *ADH1B* genotypes [1–6]. Current flushers were less frequently reported in the subject groups with alcohol dependence than in the control group, and never flushers and former flushers showed similarly elevated ORs for alcohol dependence as compared to current flushers, among both men and women. However, the flushing status subdivided subjects with each of the *ALDH2* and *ADH1B* genotypes into groups with substantially lower and higher ORs. Never or former flushing markedly increased the ORs of alcohol dependence among carriers of any *ALDH2* and *ADH1B* genotype combination, compared with current flushing. Commercially available *ALDH2* plus *ADH1B* genotyping has been used as a preventive tool against alcohol dependence in Japan. Combined evaluation of the alcohol flushing status and

**Table 6. Temporal profiles for drinking and flushing in former-flushing, alcohol-dependent subjects according to *ALDH2* genotype.**

| | Total | *ALDH2* genotype | | |
|---|---|---|---|---|
| | | *1/*1 (active) | *1/*2 + *2/*2 (inactive) | P |
| Number of patients (61 men and 5 women) | 66 | 28 | 38 | |
| Age (years) | 53.9±10.9 | 53.4±11.4 | 54.3±10.5 | 0.72 |
| Age at the first hospital visit for alcohol-related problems[a] | 51.4±12.4 | 50.9±13.6 | 51.8±11.5 | 0.77 |
| Age at the start of drinking (years) | 18.8±4.3 | 18.9±3.9 | 18.7±4.7 | 0.87 |
| Age up until which the flushing tendency continued (years) | 25.7±6.1 | 24.6±7.2 | 26.5±5.1 | 0.24 |
| Duration of flushing tendency (years) | 8.0±5.9 | 6.8±6.5 | 8.8±5.4 | 0.16 |
| Age at the start of habitual drinking (years) | 25.9±7.4 | 25.4±8.3 | 26.2±6.7 | 0.23 |
| Disappearance of flushing tendency | | | | |
| Before the start of habitual drinking | 45.5% | 42.9% | 47.4% | 0.91 |
| Within 5 years after the start of habitual drinking | 36.4% | 39.3% | 34.2% | |
| More than 6 years after the start of habitual drinking | 18.2% | 17.9% | 18.4% | |

*ALDH2*, aldehyde dehydrogenase-2. Data are expressed as the mean ± SD and percentage values. N = 1 for *ALDH2*2/*2*.

[a] Alcohol-related organ injuries or alcohol dependence.

the *ALDH2* plus *ADH1B* genotype may show superior ability than that of either alone for predicting the risk of alcohol dependence.

The temporal profiles of drinking and flushing were evaluated, for the first time, in former flushers with alcohol dependence; these profiles revealed that the flushing tendency disappeared soon after or before the start of habitual drinking, regardless of the *ALDH2* genotype. These results suggest that constitutional or acquired tolerance against alcohol flushing after a small amount of drinking is another independent susceptibility trait for alcohol dependence. Tolerance against alcohol flushing may be acquired due to a pattern of heavy alcohol use in some subjects during their young adulthood, and disappearance of flushing in some former flushers could have been a consequence of alcohol dependence rather than a risk factor for alcohol dependence. It would be more relevant to study detailed drinking behavior during the young adulthood. However, the flushing tendency disappeared before the start of habitual drinking in 45.5% of former flushers, and this interpretation was not applicable to never flushers among inactive *ALDH2*\*1/\*2 carriers. The mean age until which the flushing tendency continued and the mean age at the first hospital visit for alcohol-related problems were 25.7 years and 51.4 years, respectively. Thus, use of alcohol flushing data in addition to *ALDH2* genotyping is useful to predict, and could be expected to reduce, the risk of future development of alcohol dependence, especially in younger generations. Combined use of this information with established screening tools, e.g., Alcohol Use Disorders Identification Test (AUDIT; [18]), may be more useful to detect alcohol use disorders that are already present.

It was also puzzling that there were no differences in the temporal profiles of the disappearance of alcohol flushing according to the *ALDH2* genotypes. These results suggest the presence of other unidentified factors that prevent the alcohol flushing tendency. A cutaneous flushing, increase in skin temperature, decrease in blood pressure after drinking have been reported to be significantly blocked by the administration of antihistamine H1 and H2 receptor antagonists in Asian flushers [19–22]. Aspirin [23] and opioid antagonist, nalmefene [24], also weaken some flushing responses in Asian flushers. As well as the pathways via the central and peripheral histamine, prostaglandins, and opioid receptors, several cascades from the triggers of ethanol and acetaldehyde to the final flushing responses may participate in the flushing responses. Further evaluation of the background of the never or former flushing trait is warranted in future research, combining flushing-related genomics and drinking pattern in young adulthood.

Inactive ALDH2 has been reported as a strong risk factor for squamous cell carcinoma of the upper aerodigestive tract in East Asian drinkers [25, 26]. Current or former flushing has been shown to be a predictor of the risk of cancer as a surrogate marker of inactive ALDH2 in various drinking populations [27]. However, the cancer risk in the current or former flushers was substantially lower than the risk in *ALDH2*\*1/\*2 carriers in an alcohol-dependent population overall [27]. This could be partly attributable to the existence of additional causes of alcohol flushing besides the *ALDH2* genotype in persons susceptible to alcohol dependence. Investigation is warranted, in a future study, of whether and how the combination of alcohol flushing status and the *ALDH2* plus *ADH1B* genotype might affect drinking behaviors and manifestations of alcohol-related problems.

The present study had several limitations. This case control study was based on the results of questions about facial flushing tendency after a glass of beer. Recall bias of the responses to the questionnaire cannot be ruled out, and the results regarding associations of alcohol flushing with *ALDH2* genotype and susceptibility to alcohol dependence differ according to how the question about alcohol flushing is posed. When Japanese subjects were asked about current facial flushing after drinking alcohol without specifying the alcohol dose, half of those with

active ALDH2 were sometimes or always flushers, because they experienced facial flushing after drinking a substantial amount of alcohol [28], and the sometimes flushers showed the highest risk for alcohol abuse [29]. However, when current or former flushing individuals after a glass of beer in the present questionnaire were assumed to have the inactive *ALDH2*2* allele, alcohol flushing was found to serve as a surrogate marker of the *ALDH2*2* allele carriers with a sensitivity and specificity of approximately 90% in the Japanese general population [12, 15, 30]. In Japan, it is customary to toast each other with a glass ($\approx$180 mL) of beer at social gatherings. Cultural differences, including popular alcoholic beverage types and typical sizes of beer glasses, could influence the results of the flushing questionnaire. The sensitivity and specificity of the questionnaire have been previously reported as 79%-95% and 77%-82%, respectively, in Korea [31] and 89% and 81%, respectively, in Taiwan [32]. Caucasians who reported flushing responses after 1 or 2 drink, which is equivalent to approximately 2 or 3 typical Japanese glasses of beer, were more likely to have symptoms of alcohol dependence, and report a parental history of alcohol problems [33]. Such Caucasians' flushing after moderate amounts of alcohol may be a risk factor in contrast with the protective role of the Japanese flushing after a small amount of alcohol. An oral alcohol challenge test using 0.56 g ethanol/kg showed lower specificity of self-reported flushing for predicting the presence of inactive ALDH2 in comparison to investigator-observed flushing [34]. Self-reported flushing may be a net self-evaluation of the perception of general discomfort [35] and cardiovascular effects [36] in flushing responses.

Another limitation was that alcohol dependence had not been excluded from the historical controls who had registered in case control studies of esophageal cancer as cancer-free controls between 2000 and 2001 [12, 15, 37]. The proportion of control subjects who drank alcohol on 3 or more days per week and consumed at least 22 g of ethanol at a time was 56.0% in men [37] and 12.9% in women [15]. These values were similar to the 53.3% in male adults and 9.1% in female adults reported by the National Nutrition Survey in 2001 [38]. A 2003 nationwide survey showed a low prevalence (1.5% in men and 0.2% in women) of lifetime alcohol dependence diagnosed according to the ICD-10 criteria in Japanese adults [11]. The mortality rate associated with esophageal cancer per 100,000 in 2001 was 10.5 in men and 1.3 in women [39]. Thus, although the *ADH1B* and *ALDH2* genotypes are risk factors for both alcohol dependence [6–10] and esophageal cancer [15, 25, 37], it is unlikely that the potential selection bias of the controls, i.e., esophageal-cancer-free and imperfect exclusion of alcohol dependence, significantly affected the present results due to their low prevalence.

Although the total number of alcohol-dependent patients was large, the numbers in some categories of combinations of alcohol flushing status and genotypes were small, especially for the female patients and the control groups. In addition to the retrospective self-reported alcohol flushing questionnaire, the temporal pattern of former flushing also has potential recall bias. Furthermore, it was evaluated mainly in male subjects with alcohol dependence, which may differ from the case in the general population and women. There is a higher probability of Chinese university students thinking that they should encourage a flusher to stop drinking or drink less when the flusher is a woman [40], which may prevent the development of tolerance in female flushers. Generalization of the results obtained in this study, which was based on investigations of treatment-seeking alcohol-dependent patients, will require confirmation in various heavy-drinking populations, including groups with mild alcohol-dependent cases and a large number female cases, as well as in large control populations.

In conclusion, never flushing and former flushing were identified as strong independent risk factors for alcohol dependence in carriers of any *ALDH2* and *ADH1B* genotypes, and combined evaluation of the self-reported alcohol flushing history and the *ALDH2* and *ADH1B* genotypes might enable better estimation of the risk of alcohol dependence in Asians.

## Supporting information

**S1 Data.**
(XLSX)

## Author Contributions

**Conceptualization:** Akira Yokoyama.

**Data curation:** Akira Yokoyama, Tetsuji Yokoyama, Mitsuru Kimura, Sachio Matsushita, Masako Yokoyama.

**Formal analysis:** Tetsuji Yokoyama.

**Investigation:** Akira Yokoyama, Mitsuru Kimura, Sachio Matsushita, Masako Yokoyama.

**Methodology:** Akira Yokoyama, Tetsuji Yokoyama.

**Project administration:** Akira Yokoyama.

**Writing – original draft:** Akira Yokoyama, Tetsuji Yokoyama.

**Writing – review & editing:** Akira Yokoyama, Tetsuji Yokoyama, Mitsuru Kimura, Sachio Matsushita, Masako Yokoyama.

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
