## [Decision Letter · Decision Letter 0]

9 Jun 2021

PONE-D-21-07074

Combinations of alcohol-induced flushing with genetic polymorphisms of alcohol and aldehyde dehydrogenases and the risk of alcohol dependence in Japanese men and women

PLOS ONE

Dear Dr. Yokoyama,

Thank you for submitting your manuscript to PLOS ONE. After careful consideration, we feel that it has merit but does not fully meet PLOS ONE’s publication criteria as it currently stands. Therefore, we invite you to submit a revised version of the manuscript that addresses the points raised during the review process.

Importantly pleae indicate more clearly which restrictions apply for your data sharing. **PLOS journals require authors to make all data necessary to replicate their study’s findings publicly available without restriction at the time of publication. When specific legal or ethical restrictions prohibit public sharing of a data set, authors must indicate how others may obtain access to the data.**

Authors must share the** “minimal data set”** for their submission. PLOS defines the minimal data set to consist of the data required to replicate all study findings reported in the article, as well as related metadata and methods. Additionally, you may consider deposition of data in a field relevant repository.

Once more we apologize for the delay in the evaluation process, but are confident that your manuscript can be accepted for publication when you have improved your data sharing and after you have implemented corrections as suggested by the reviewers.

We look forward to receiving your revised manuscript.

Kind regards,

Nadine Bernhardt, Ph.D.

Academic Editor

PLOS ONE

3. Please include your tables as part of your main manuscript and remove the individual files. Please note that supplementary tables should  be uploaded) as separate "supporting information" files.

Reviewers' comments:

Reviewer's Responses to Questions

**Comments to the Author**

1. Is the manuscript technically sound, and do the data support the conclusions?

Reviewer #1: Partly

Reviewer #2: Yes

2. Has the statistical analysis been performed appropriately and rigorously? 

Reviewer #1: Yes

Reviewer #2: Yes

3. Have the authors made all data underlying the findings in their manuscript fully available?

Reviewer #1: No

Reviewer #2: Yes

4. Is the manuscript presented in an intelligible fashion and written in standard English?

Reviewer #1: Yes

Reviewer #2: Yes

5. Review Comments to the Author

Reviewer #1: This manuscript examined ALDH2/ADH1B genotypes and alcohol flushing using a case-control study of alcohol use disorder (AUD) in over 5,000 Japanese men and women. Findings suggest that a two-item measure of past and current self-reported alcohol flushing was independently associated with AUD, over and above well-documented ALDH2 and ADH1B susceptibility genotypes.

Strengths of this study include its innovative approach disaggregating former flushing from ALDH2/ADH1B genotypes, use of both men and women, a relatively large sample size, and double-blind administration of the alcohol flushing questionnaire. Notwithstanding these strengths, there were also several significant concerns that dampened my enthusiasm for this project.

1. My most significant concern relates to potential control selection bias. First, individuals with AUD were not excluded from the control group. Although the authors estimate AUD rates may be low based on similar prior samples, the number of ‘AUD controls’ who have AUD and the impact of imperfect classification here ultimately remains unknown. Second, there was no data provided on alcohol use levels in controls. The ability to report on flushing and the hypothesized mechanisms of ALDH2/ADH1B on AUD all require alcohol consumption. If available, any data on alcohol use within controls would help estimate the strength of these concerns. Third, controls were borrowed from an esophageal cancer study. ALDH2/ADH1B genotypes are a known risk factor for esophageal cancer, such that it is less likely these cancer-free controls possessed certain genotypes. More attention to these limitations is necessary, with a much more nuanced and cautious interpretation of study findings.

2. The authors advocate for use of alcohol flushing data in addition to genotype to predict risk (rather than presence) of AUD. This was conceptually unclear and at times contradictory. The authors discuss how flushing might be present early on in individuals’ drinking histories and then disappear around the time habitual drinking emerges. They also discuss former flushing as a potential correlate of tolerance, a symptom of AUD. These interpretations seem to support former flushing as a correlate, rather than as a risk factor. Greater clarity is needed, especially because this could impact clinical implications (eg, whether flushing questions are useful to predict those who will go on to develop AUD or to use as screeners to detect AUD already present).

3. More information is needed to evaluate methodological quality of the alcohol flushing data: (i) Are data available on the measure’s reliability and validity? (ii) Has the self-reported questionnaire been correlated to observed flushing in the laboratory? (doi:10.15288/jsa.1996.57.267 notes differences in self-report and observed flushing) (iii) Have any psychometrics been reported in women? The authors do present some data on sensitivity/specificity in the limitations, but these are for detecting ALDH2 genotypes rather than capturing actual flushing response or flushing’s relation to AUD.

Additional concerns:

1. References are needed for lines 82-86.

2. It was unclear how the price of ALDH2/ADH1B genotyping is an argument in support of the current study (lines 86-88). The authors suggest genotype + flushing data better predicts AUD than flushing data alone, so no genotyping costs would be saved in this screening approach.

3. The introduction would benefit from discussing how knowledge about timing of flushing disappearance may help understand relations of alcohol flushing and ALDH2/ADH1B with AUD. Also, any prior findings on timing of flushing disappearance should be introduced.

4. Specify any a priori hypotheses in the final paragraph of the introduction.

5. There was a large amount of missing data on the temporal patterning of former flushing. This appears to be partly due to the item being added to assessments more recently. The authors should make this more explicit in the text if correct, report any other reasons for missing data, and explain how missing data were handled in analyses. This restricted sample may be less representative depending on the reasons for missing data, and this should be added as a study limitation. The authors could test and report any differences in study variables and demographics as a function of those with and without missing data to speak to generalizability concerns.

6. The authors should note in the text whether any part of this sample have been previously published on and, if so, include relevant citations and the need for these analyses.

7. The manuscript would benefit from an expanded discussion on what these findings mean for the broader literature on ALDH2/ADH1B relationships with alcohol flushing. Prior research has tended to use flushing questionnaires to infer ALDH2 genotype and risk for alcohol and cancer outcomes. The current findings suggest additional sources of variability in alcohol flushing aside from the ALDH2 and ADH1B genotypes tested. If alcohol flushing is not fully due to ALDH2/ADH1B, what implications does this have for interpreting existing literature, for using flushing questionnaires to infer genotypes, and for identifying AUD?

8. The discussion should mention limitations to the temporal patterning of former flushing data, in addition it the current discussion on the flushing questionnaire. There is potential recall bias for this retrospective self-report measure, with possible impacts on strength/direction of results.

9. There were several instances in which the text referred to men rather than men and women, and these should be revised as appropriate (line 100, line 238, Table 6 title).

10. Minor typo: The percentage listed on line 171 seems to contradict Table 1.

Reviewer #2: Alcohol-induced facial flushing and genetic polymorphisms of alcohol dehydrogenase (the fast-metabolizing ADH1B, rs122998) and aldehyde dehydrogenases (the inactive ALDH2, rs671), which enhance alcohol-induced flushing, are known protective factors for alcohol dependence (AD) in Asians. In this study, the authors examined whether combination of the flushing status and the 2 ADH1B/ALDH2 genotypes could better predict the risk of AD in Japanese men and women. Their study cohorts included a large number of male (3721) and a much smaller number female (335) AD cases and 610 male and 406 female historical community controls. The alcoholic and control subjects included current, former, and never flushing individuals as defined by simple questionnaires. They showed that never or former alcohol flushing, the slow-metabolizing ADH1B allele and the active ALDH2 genotype were risk factors for AD. In addition, never or former flushing and the ADH1B/ALDH2 genotype combinations were independent risk factors of AD in both men and women. Consistent with the finding that a large portion of the never or former flushing AD and control subjects were found to carry the inactive ALDH2 genotype, they also showed that non-flushing increased the OR of AD regardless of the ALDH2/ADH1B genetic polymorphisms. In a separate group of 66 former flushing AD, flushing response was shown to disappear around the start of alcohol misuse during early adulthood regardless of the the active and inactive ALDH2 genotype.

As the OR for the combination of the flushing status and the ALDH2/ADH1B genotype were lower than the OR for the flushing status alone or the genotypes alone, it is not clear how the combination can provide a better new strategy for AD risk assessment as claimed by the authors. Other than that, the study was well done and well written. There are no other major concerns. The following are some minor concerns and questions.

In line 171, the numbers 7.0% and 4.0% are different from those in Table 1.

Table 1 needs to show the p values for all the comparisons. For former flushing, was the differences significant between the AD and controls?

For Figure 1 and Figure 2, the different results as to the ADH1B genotype between males and females likely was due to a much small number of female subjects.

There is a discrepancy in the gender description of the subjects in Table 6. The title says it is for AD men. However, the Methods indicated that the 66 subjects include 61 men and 5 women.

6. PLOS authors have the option to publish the peer review history of their article (what does this mean?). If published, this will include your full peer review and any attached files.

Reviewer #1: No

Reviewer #2: No

---

## [Author Response · Author response to Decision Letter 0]

7 Jul 2021

Prof. Nadine Bernhardt

Academic Editor

PLOS ONE 　　　　　　　　　　　　　 

Dear Prof. Bernhardt:

Ref: PONE-D-21-07074

We appreciate the comments of the reviewers. They were helpful in strengthening and clarifying portions of the manuscript. 

Our point by point reply is included below:

According to the Editor’s suggestion, we share the large main data set because of its adequate anonymous nature.

Reviewer #1: This manuscript examined ALDH2/ADH1B genotypes and alcohol flushing using a case-control study of alcohol use disorder (AUD) in over 5,000 Japanese men and women. Findings suggest that a two-item measure of past and current self-reported alcohol flushing was independently associated with AUD, over and above well-documented ALDH2 and ADH1B susceptibility genotypes.

Strengths of this study include its innovative approach disaggregating former flushing from ALDH2/ADH1B genotypes, use of both men and women, a relatively large sample size, and double-blind administration of the alcohol flushing questionnaire. Notwithstanding these strengths, there were also several significant concerns that dampened my enthusiasm for this project.

1. My most significant concern relates to potential control selection bias. First, individuals with AUD were not excluded from the control group. Although the authors estimate AUD rates may be low based on similar prior samples, the number of ‘AUD controls’ who have AUD and the impact of imperfect classification here ultimately remains unknown. Second, there was no data provided on alcohol use levels in controls. The ability to report on flushing and the hypothesized mechanisms of ALDH2/ADH1B on AUD all require alcohol consumption. If available, any data on alcohol use within controls would help estimate the strength of these concerns. Third, controls were borrowed from an esophageal cancer study. ALDH2/ADH1B genotypes are a known risk factor for esophageal cancer, such that it is less likely these cancer-free controls possessed certain genotypes. More attention to these limitations is necessary, with a much more nuanced and cautious interpretation of study findings.

Reply:　According to the reviewer’s comment, we added the following discussion.

Page 17, line 327: Another limitation was that alcohol dependence had not been excluded from the historical controls who had registered in case control studies of esophageal cancer as cancer-free controls between 2000 and 2001 (12,15,37). The proportion of control subjects who drank alcohol on 3 or more days per week and consumed at least 22 g of ethanol at a time was 56.0% in men (37) and 12.9% in women (15). These values were similar to the 53.3% in male adults and 9.1% in female adults reported by the National Nutrition Survey in 2001 (38). A 2003 nationwide survey showed a low prevalence (1.5% in men and 0.2% in women) of lifetime alcohol dependence diagnosed according to the ICD-10 criteria in Japanese adults (11). The mortality rate associated with esophageal cancer per 100,000 in 2001 was 10.5 in men and 1.3 in women (39). Thus, although the ADH1B and ALDH2 genotypes are risk factors for both alcohol dependence (6-10) and esophageal cancer (15,25,37), it is unlikely that the potential selection bias of the controls, i.e., esophageal-cancer-free and imperfect exclusion of alcohol dependence, significantly affected the present results due to their low prevalence.

2. The authors advocate for use of alcohol flushing data in addition to genotype to predict risk (rather than presence) of AUD. This was conceptually unclear and at times contradictory. The authors discuss how flushing might be present early on in individuals’ drinking histories and then disappear around the time habitual drinking emerges. They also discuss former flushing as a potential correlate of tolerance, a symptom of AUD. These interpretations seem to support former flushing as a correlate, rather than as a risk factor. Greater clarity is needed, especially because this could impact clinical implications (eg, whether flushing questions are useful to predict those who will go on to develop AUD or to use as screeners to detect AUD already present).

Reply: We discussed these issues as follows.

Page 15, line 260: Tolerance against alcohol flushing may be acquired due to a pattern of heavy alcohol use in some subjects during their young adulthood, and disappearance of flushing in some former flushers could have been a consequence of alcohol dependence rather than a risk factor for alcohol dependence. It would be more relevant to study detailed drinking behavior during the young adulthood. However, the flushing tendency disappeared before the start of habitual drinking in 45.5% of former flushers, and this interpretation was not applicable to never flushers among inactive ALDH2*1/*2 carriers. The mean age until which the flushing tendency continued and the mean age at the first hospital visit for alcohol-related problems were 25.7 years and 51.4 years, respectively. Thus, use of alcohol flushing data in addition to ALDH2 genotyping is useful to predict, and could be expected to reduce, the risk of future development of alcohol dependence, especially in younger generations. Combined use of this information with established screening tools, e.g., Alcohol Use Disorders Identification Test (AUDIT; 18), may be more useful to detect alcohol use disorders that are already present.

3. More information is needed to evaluate methodological quality of the alcohol flushing data: (i) Are data available on the measure’s reliability and validity? (ii) Has the self-reported questionnaire been correlated to observed flushing in the laboratory? (doi:10.15288/jsa.1996.57.267 notes differences in self-report and observed flushing) (iii) Have any psychometrics been reported in women? The authors do present some data on sensitivity/specificity in the limitations, but these are for detecting ALDH2 genotypes rather than capturing actual flushing response or flushing’s relation to AUD.

Reply: We discussed these issues as follows.

Page 17, line 322: An oral alcohol challenge test using 0.56 g ethanol/kg showed lower specificity of self-reported flushing for predicting the presence of inactive ALDH2 in comparison to investigator-observed flushing (34). Self-reported flushing may be a net self-evaluation of the perception of general discomfort (35) and cardiovascular effects (36) in flushing responses.

Additional concerns:

1. References are needed for lines 82-86.

Reply: We added the reference (6).

2. It was unclear how the price of ALDH2/ADH1B genotyping is an argument in support of the current study (lines 86-88). The authors suggest genotype + flushing data better predicts AUD than flushing data alone, so no genotyping costs would be saved in this screening approach.

Reply: We changed the sentence as follows. 

Page 5, line 77: ALDH2 plus ADH1B genotyping using buccal smear DNA is commercially available and inexpensive in Japan.

3. The introduction would benefit from discussing how knowledge about timing of flushing disappearance may help understand relations of alcohol flushing and ALDH2/ADH1B with AUD. Also, any prior findings on timing of flushing disappearance should be introduced. 4. Specify any a priori hypotheses in the final paragraph of the introduction.

Reply: According to the reviewer’ comment, we added the following paragraph before the final paragraph of the introduction.

Page 6, line 97: According to a report, alcohol consumption in inactive ALDH2*1/*2 carriers in the general population increased in the order of current flushers, former flushers, and never flushers (12). If the flushing disappearance occurred in young adulthood and not around the time of development of alcohol dependence in many former flushers who are alcohol-dependent, it is conceivable that both former flushing and never flushing may serve as independent risk factors for alcohol dependence, in combination with the ADH1B and ALDH2 genotypes.

5. There was a large amount of missing data on the temporal patterning of former flushing. This appears to be partly due to the item being added to assessments more recently. The authors should make this more explicit in the text if correct, report any other reasons for missing data, and explain how missing data were handled in analyses. This restricted sample may be less representative depending on the reasons for missing data, and this should be added as a study limitation. The authors could test and report any differences in study variables and demographics as a function of those with and without missing data to speak to generalizability concerns.

Reply: To clarify that data of the temporal patterning of former flushing were representative data in the population, we added how the data were corrected in the methods.

Page 9, line 135: Between July 2019 and December 2020, 283 Japanese subjects with alcohol dependence (253 men and 30 women) aged 40–79 years who visited the Center for the first time were asked about their alcohol flushing history and underwent routine ADH1B and ALDH2 genotyping. Of the subjects, 7 (6 men and one women), 69 (64 men and 5 women), and 207 (183 men and 24 women) reported themselves as current flushers, former flushers, and never flushers, respectively. We asked 66 former flushers (61 men and 5 women) the following question: “Up until what age did you have a tendency to develop facial flushing immediately after drinking a glass of beer?”

6. The authors should note in the text whether any part of this sample have been previously published on and, if so, include relevant citations and the need for these analyses.

Reply: According to the comment, we added the following sentence in the methods.

Page 7, line 119: Male subjects with alcohol dependence were a subgroup (40-79 years) of 4051 subjects aged 30-79 years, in which we recently reported the associations between the results of an alcohol flushing questionnaire and the ALDH1B and ALDH2 genotypes (6).

7. The manuscript would benefit from an expanded discussion on what these findings mean for the broader literature on ALDH2/ADH1B relationships with alcohol flushing. Prior research has tended to use flushing questionnaires to infer ALDH2 genotype and risk for alcohol and cancer outcomes. The current findings suggest additional sources of variability in alcohol flushing aside from the ALDH2 and ADH1B genotypes tested. If alcohol flushing is not fully due to ALDH2/ADH1B, what implications does this have for interpreting existing literature, for using flushing questionnaires to infer genotypes, and for identifying AUD?

Reply: We discussed these issue as follows in the discussion.

Page 16, line 288: Inactive ALDH2 has been reported as a strong risk factor for squamous cell carcinoma of the upper aerodigestive tract in East Asian drinkers (25,26). Current or former flushing has been shown to be a predictor of the risk of cancer as a surrogate marker of inactive ALDH2 in various drinking populations (27). However, the cancer risk in the current or former flushers was substantially lower than the risk in ALDH2*1/*2 carriers in an alcohol-dependent population overall (27). This could be partly attributable to the existence of additional causes of alcohol flushing besides the ALDH2 genotype in persons susceptible to alcohol dependence. Investigation is warranted, in a future study, of whether and how the combination of alcohol flushing status and the ALDH2 plus ADH1B genotype might affect drinking behaviors and manifestations of alcohol-related problems.

8. The discussion should mention limitations to the temporal patterning of former flushing data, in addition it the current discussion on the flushing questionnaire. There is potential recall bias for this retrospective self-report measure, with possible impacts on strength/direction of results.

Reply: We discussed these issues as follows in the discussion.

Page 18, line 345: In addition to the retrospective self-reported alcohol flushing questionnaire, the temporal pattern of former flushing also has potential recall bias. Furthermore, it was evaluated only in male subjects with alcohol dependence, which may differ from the case in the general population and women. There is a higher probability of Chinese university students thinking that they should encourage a flusher to stop drinking or drink less when the flusher is a woman (40), which may prevent the development of tolerance in female flushers. 

9. There were several instances in which the text referred to men rather than men and women, and these should be revised as appropriate (line 100, line 238, Table 6 title).

Reply: According this comments, we checked and correct the text appropriately. 

10. Minor typo: The percentage listed on line 171 seems to contradict Table 1.

Reply:　We corrected the percentage.

Reviewer #2: Alcohol-induced facial flushing and genetic polymorphisms of alcohol dehydrogenase (the fast-metabolizing ADH1B, rs122998) and aldehyde dehydrogenases (the inactive ALDH2, rs671), which enhance alcohol-induced flushing, are known protective factors for alcohol dependence (AD) in Asians. In this study, the authors examined whether combination of the flushing status and the 2 ADH1B/ALDH2 genotypes could better predict the risk of AD in Japanese men and women. Their study cohorts included a large number of male (3721) and a much smaller number female (335) AD cases and 610 male and 406 female historical community controls. The alcoholic and control subjects included current, former, and never flushing individuals as defined by simple questionnaires. They showed that never or former alcohol flushing, the slow-metabolizing ADH1B allele and the active ALDH2 genotype were risk factors for AD. In addition, never or former flushing and the ADH1B/ALDH2 genotype combinations were independent risk factors of AD in both men and women. Consistent with the finding that a large portion of the never or former flushing AD and control subjects were found to carry the inactive ALDH2 genotype, they also showed that non-flushing increased the OR of AD regardless of the ALDH2/ADH1B genetic polymorphisms. In a separate group of 66 former flushing AD, flushing response was shown to disappear around the start of alcohol misuse during early adulthood regardless of the the active and inactive ALDH2 genotype.

As the OR for the combination of the flushing status and the ALDH2/ADH1B genotype were lower than the OR for the flushing status alone or the genotypes alone, it is not clear how the combination can provide a better new strategy for AD risk assessment as claimed by the authors. Other than that, the study was well done and well written. There are no other major concerns. 

Reply: To clarify these issues, we added the following discussion.

Page 14, line 243: Current flushers were less frequently reported in the subject groups with alcohol dependence than in the control group, and never flushers and former flushers showed similarly elevated ORs for alcohol dependence as compared to current flushers, among both men and women. However, the flushing status subdivided subjects with each of the ALDH2 and ADH1B genotypes into groups with substantially lower and higher ORs. Never or former flushing markedly increased the ORs of alcohol dependence among carriers of any ALDH2 and ADH1B genotype combination, compared with current flushing. Commercially available ALDH2 plus ADH1B genotyping has been used as a preventive tool against alcohol dependence in Japan. Combined evaluation of the alcohol flushing status and the ALDH2 plus ADH1B genotype may show superior ability than that of either alone for predicting the risk of alcohol dependence.

The following are some minor concerns and questions.

In line 171, the numbers 7.0% and 4.0% are different from those in Table 1.

Reply: We corrected the numbers correctly.

Table 1 needs to show the p values for all the comparisons. For former flushing, was the differences significant between the AD and controls?

Reply: We added the results of comparing the prevalence of each category between alcohol dependence and controls in the table 1.

For Figure 1 and Figure 2, the different results as to the ADH1B genotype between males and females likely was due to a much small number of female subjects.

Reply: We need to evaluate the possible gender difference and discussed this issue as follows.

Page 18, line 347: Furthermore, it was evaluated only in male subjects with alcohol dependence, which may differ from the case in the general population and women. There is a higher probability of Chinese university students thinking that they should encourage a flusher to stop drinking or drink less when the flusher is a woman (40), which may prevent the development of tolerance in female flushers. Generalization of the results obtained in this study, which was based on investigations of treatment-seeking alcohol-dependent patients, will require confirmation in various heavy-drinking populations, including groups with mild alcohol-dependent cases and a large number female cases, as well as in large control populations.

There is a discrepancy in the gender description of the subjects in Table 6. The title says it is for AD men. However, the Methods indicated that the 66 subjects include 61 men and 5 women.

Reply: We corrected the title appropriately.

---

## [Decision Letter · Decision Letter 1]

14 Jul 2021

Combinations of alcohol-induced flushing with genetic polymorphisms of alcohol and aldehyde dehydrogenases and the risk of alcohol dependence in Japanese men and women

PONE-D-21-07074R1

Dear Dr. Yokoyama,

We appreciate the effort you have taken to address all issues raised during the revision process and thus we’re pleased to inform you that your manuscript has been judged scientifically suitable for publication and will be formally accepted for publication once it meets all outstanding technical requirements.

Kind regards,

Nadine Bernhardt, Ph.D.

Academic Editor

PLOS ONE

Reviewers' comments:

Reviewer's Responses to Questions

**Comments to the Author**

1. If the authors have adequately addressed your comments raised in a previous round of review and you feel that this manuscript is now acceptable for publication, you may indicate that here to bypass the “Comments to the Author” section, enter your conflict of interest statement in the “Confidential to Editor” section, and submit your "Accept" recommendation.

Reviewer #1: (No Response)

Reviewer #2: All comments have been addressed

2. Is the manuscript technically sound, and do the data support the conclusions?

Reviewer #1: Yes

Reviewer #2: Yes

3. Has the statistical analysis been performed appropriately and rigorously? 

Reviewer #1: Yes

Reviewer #2: Yes

4. Have the authors made all data underlying the findings in their manuscript fully available?

Reviewer #1: Yes

Reviewer #2: Yes

5. Is the manuscript presented in an intelligible fashion and written in standard English?

Reviewer #1: Yes

Reviewer #2: Yes

6. Review Comments to the Author

Reviewer #1: The authors were generally responsive to my comments, and my opinions on the strengths of this study remain the same.

My most significant prior concern was potential control selection bias. The authors added data on alcohol use within controls, which was generally comparable to a past national survey, and I think this strengthens interpretability of the results. Although the authors are still limited by available data such that misclassification (AUD in controls) and uneven exposure (controls who have not consumed alcohol) remain unknown, I appreciate their work estimating comparability and willingness to speak directly to these challenges in the text.

Reviewer #2: (No Response)

7. PLOS authors have the option to publish the peer review history of their article (what does this mean?). If published, this will include your full peer review and any attached files.

Reviewer #1: No

Reviewer #2: No

---

## [Editor Report · Acceptance letter]

16 Jul 2021

PONE-D-21-07074R1 

Combinations of alcohol-induced flushing with genetic polymorphisms of alcohol and aldehyde dehydrogenases and the risk of alcohol dependence in Japanese men and women 

Dear Dr. Yokoyama:

I'm pleased to inform you that your manuscript has been deemed suitable for publication in PLOS ONE. Congratulations! Your manuscript is now with our production department. 

Kind regards, 

on behalf of

Dr. Nadine Bernhardt 

Academic Editor

PLOS ONE